

# Combined spatial and frequency dual stream network for face forgery detection

Hui Zhao[1,2,*], Xin Li[1,2,*], Bingxin Xu[1,2] and Hongzhe Liu[1,2]

[1] Department of Robotics, Beijing Union University, Beijing, China
[2] Beijing Key Laboratory of Information Service Engineering, Beijing Union University, Beijing, China
* These authors contributed equally to this work.

## ABSTRACT

With the development of generative model, the cost of facial manipulation and forgery is becoming lower and lower. Fraudulent data has brought numerous hidden threats in politics, privacy, and cybersecurity. Although many methods of face forgery detection focus on the learning of high frequency forgery traces and achieve promising performance, these methods usually learn features in spatial and frequency independently. In order to combine the information of the two domains, a combined spatial and frequency dual stream network is proposed for face forgery detection. Concretely, a cross self-attention (CSA) module is designed to improve frequency feature interaction and fusion at different scales. Moreover, to augment the semantic and contextual information, frequency guided spatial feature extraction module is proposed to extract and reconstruct the spatial information. These two modules deeply mine the forgery traces *via* a dual-stream collaborative network. Through comprehensive experiments on different datasets, we demonstrate the effectiveness of proposed method for both within and cross datasets.

## INTRODUCTION

In the development process of image generation technology, the early variational autoencoder (VAE) (*Kingma & Welling, 2013*) and the generative adversarial network (GAN) (*Creswell et al., 2018*) laid the foundation for the deepfake technology. Nowadays, StyleGAN (*Karras, Laine & Aila, 2019*) and ProGAN (*Gao, Pei & Huang, 2019*) models are capable of synthesizing high-quality generated images. In recent years, the diffusion model (*Ho, Jain & Abbeel, 2020*) has brought the image generation technology to a new height, followed by the trust crisis in financial, political, and other fields caused by generated information. Therefore, constructing an effective and accurate face forgery detection method is of great practical significance.

At present, deep learning based face forgery detection methods can be roughly divided into two categories, namely spatial feature based detection methods and frequency based detection methods. The first type of methods attempt to extract features from image's spatial pixels and further investigate forgery traces in the generated image, such as texture differences generated during local forgery (*Zhao et al., 2021a*; *Liu, Qi & Torr, 2020*) and

Corresponding authors
Bingxin Xu, xubingxin@buu.edu.cn
Hongzhe Liu, liuhongzhe@buu.edu.cn

inconsistencies in the sources of different forgery regions (*Ju et al., 2022*; *Zhao et al., 2021b*). These observable differences are gradually rectified as generative technology progress. These methods solely focus on high-level semantic information in the spatial domain and ignore low-level signal variations in forged images, limiting their robustness and effectiveness. This also motivates researches into frequency domain based face forgery detection (*Qian et al., 2020*; *Li et al., 2021*). In terms of frequency feature extraction, researchers often use several kinds of frequency transformation methods to extract manipulated traces hidden in forgery images. However, some methods tend to become extremely complex and suffer from parameter redundancy, making it difficult to learn simple and useful features. Therefore, how to properly extract invisible frequency features is a pressing issue that must be addressed. To achieve a proper integration of spatial and frequency features, some dual-stream networks (*Gu et al., 2022*; *Shuai et al., 2023*; *Li et al., 2022*) have been proposed in the existing studies. These methodologies duly recognize the inherent synergy between frequency and spatial features, endeavoring to foster their interactive learning by means of attention mechanisms or mid-level fusion techniques. Nonetheless, a crucial aspect often overlooked in these investigations pertains to the distinct characteristics exhibited by various frequency bands within the frequency domain. In *Luo et al. (2021)*, pointed out that the information contained in high frequency reveals imperceptible artifacts, which helps to better distinguish between true and manipulated faces. Hence, we endeavor to decompose and select the high-frequency information, aiming to guide the model towards a more comprehensive understanding within both the spatial and frequency domains.

In this work, we propose a novel dual-stream framework that integrates both spatial and frequency features for face forgery detection. One branch of our framework incorporates a multi-scale frequency decomposition module, which is specifically designed to extract informative high-frequency cues. Notably, we discard low-frequency information and solely focus on learning from multi-scale high-frequency features. This allows us to emphasize the forgery clues in the frequency domain and suppressing the network's reliance on complex semantic information. Additionally, we introduce a novel cross self-attention (CSA) module to capture the fusion of features across different frequency scales. In the other branch, we introduce a high-frequency guided multi-scale spatial feature extraction module to extract semantic features and contextual information. Experiments demonstrate that the proposed method works well on both within-dataset and cross-dataset testing compared with other approaches.

Overall, the main contributions of this article include three aspects:

(1) We propose a multi-scale frequency feature decomposition module to efficiently capture the high-frequency clues between forgery and real images. The low-frequency components are discarded after performing the first level wavelet-packet transform. The second level wavelet-packet transform is followed to apply the high-frequency components. By combining different levels of wavelet coefficients, the model can learn more subtle variations in high-frequency;

(2) A novel cross self-attention module is proposed to efficiently integrating multi-scale frequency features while selectively emphasizing regions of interest;

(3) In the spatial feature learning branch, the inverse wavelet-packet transform (IWPT) is employed to construct a high frequency guided spatial feature extraction module. The reconstructed spatial domain can pay more attention to spatial features in different directions.

The article is organized as follows. "Related Work" provides an in-depth analysis of prior research, highlighting the limitations of existing methods, and elucidating the distinctions and advancements presented by the proposed approach. The "Proposed Method" describes comprehensive details and explanations of the proposed method. In the "Experiments" demonstrates the validity and robustness of the proposed method through extensive experiments. Finally, the "Conclusion" section encapsulates the proposed method, discussing its main strengths and weaknesses, thereby aiming to foster further exploration and advancement in this field.

## RELATED WORK

The rapid growth and popularization of face forgery technology poses security issues to facial recognition systems. To address this challenge, various methods (*Miao et al., 2023*; *Guo et al., 2023a*) for detecting face forgery have been proposed. Early methods primarily used intrinsic statistics or handcrafted features for modeling. However, these methods are labor-intensive and have poor detection performance for complex face forgery methods such as deep learning-based facial synthesis. Therefore, employing convolutional neural network (CNN)-based models to automatically learn and recognize forged attributes is a more common strategy. A type of method attempts to mine subtle artifacts from the spatial domain. Attribute network architecture is presented by *Yu, Davis & Fritz (2018)*. This model detects forged videos by using the GAN fingerprint information obtained. *Zhao et al. (2021a)* identified face forged images by using attention maps to guide the aggregation of low-level textural features and high-level semantic information. Some methods, such as face X-ray (*Li et al., 2020a*), focus on model generalization and provide an effective method for detecting mixed boundaries using self-supervised datasets. These methods, however, only mining spatial information from pixels and cannot efficiently analyze and utilize the relations between spatial information and frequency information.

Other researchers have attempted to introduce frequency information to learn the abnormal distribution in forgery images. *Li et al. (2021)* transformed the image into YUV color space and performed DCT transform to a channel attention for constructing an adaptive frequency extraction module. *Liu et al. (2022)* proposed a frequency and spatial feature fusion (FSF) module for aggregating spatial and multi-scale wavelet representations. However, forgery traces in frequency are often hidden in high frequencies. The introduction of low-frequency information often leads to model confusion. Meanwhile, these methods are coarse-grained for the use of frequency information, and thus cannot effectively select important components in high-frequency of different scales.

**Table 1 Overview of the relevant works on face forgery detection.**

| Research article | Clues | Backbone | Dataset(s) |
|---|---|---|---|
| *Miao et al. (2023)* | Frequency | F2Trans (Transformer) | FF++, Celeb-DF, DFDC, DeepFake-TIMIT-HQ |
| *Guo et al. (2023a)* | Frequency & Spatial | HiFi-Net (CNN) | LSUN, CelebaHQ, FFHQ, AFHQ, MSCOCO |
| *Yu, Davis & Fritz (2018)* | GAN fingerprint | CNN | ProGAN, SNGAN, CramerGAN, MMDGAN |
| *Zhao et al. (2021a)* | Spatial | EfficientNet-b4 (CNN) | FF++, Celeb-DF, DFDC |
| *Li et al. (2020a)* | Spatial | HRNet (CNN) | FF++, Celeb-DF, DFDC, DFD (Deep Fake Detection) |
| *Li et al. (2021)* | Frequency & Spatial | Xception (CNN) | FF++ |
| *Liu et al. (2022)* | Frequency & Spatial | Xception (CNN) | FF++, Celeb-DF, FFIW, WildDeepfake (WDF) |

Therefore, we propose a multi-scale frequency decomposition module, which effectively alleviates the mentioned issues. In order to focus on the high-frequency components at different levels, the low-frequency components in the wavelet-packet transform are removed. Additionally, a high-frequency guided spatial feature extraction module is designed for that reconstructs and analyzes multi-directional spatial features.

Table 1 provides a comprehensive overview of the relevant works on face forgery detection, consist of the utilized clues, backbone networks, and datasets considered.

# PROPOSED METHOD

## Overall architecture

Prior studies (*Xu, Zhang & Xiao, 2019*) have revealed distinctions in frequency information between forged face images and real ones. During the feature learning process, neural networks often exhibit a bias towards prioritizing low-frequency information over high-frequency counterparts, despite the potential presence of incriminating forgery clues within the latter. Inspired by this observation, we introduce a novel dual stream network that integrates spatial and frequency features. A comprehensive depiction of the network's architecture is presented in Fig. 1, which contains the following sequential steps:

(1) First, the original image is fed to the multi-scale frequency decomposition module. In order to effectively extract the high-frequency features, the first level wavelet-packet transform (WPT) is performed, followed by discarding the low-frequency components and retaining only the high-frequency components (horizontal, vertical, and diagonal) of each channel. The process is shown in Fig. 1A. In order to further capture fine-grained high-frequency features, second level wavelet packet transform is performed on the selected components and conduct selection and combination again. Details is present in Fig. 1B. Subsequently, the multi-scale high-frequency features are send to the backbone network.

(2) The CSA module is applied to promote the fusion of frequency features at the above two levels. Combining both self-attention and cross-attention mechanisms, CSA demonstrates its efficiency in promoting feature exchange and fusion across multiple scales of frequency information.

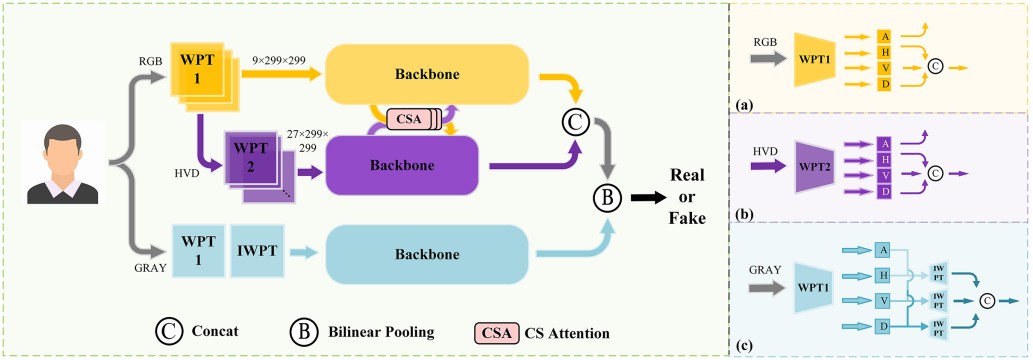

**Figure 1 The overall architecture of the proposed method.** A denotes the approximation coefficients, H denotes the horizontal coefficients, V denotes the vertical coefficients, and D denotes the diagonal coefficients.

(3) In the spatial domain branch, the original image is transformed to grayscale and processed by the multi-scale spatial information extraction module. In order to guide the reconstruction of spatial information at different scales, a standard wavelet-packet transform is applied for dividing input into four distinct frequency coefficients. Subsequently, an inverse wavelet-packet transform is employed on the various group coefficients to reconstruct spatial images. These reconstructed images are then fed into the backbone, which as presented in Fig. 1C.

(4) The bilinear pooling (*Lin, RoyChowdhury & Maji, 2015*) is utilized for merging multi-scale spatial-frequency information and sends the final features to classification heads for binary prediction.

## Multi-scale frequency decomposition module

Prior research (*Wolter et al., 2022*; *Frank et al., 2020*) has demonstrated that in various frequency ranges, forged face images have notable differences which are very helpful in identifying authenticity. Especially, as the frequency increases, the distinctions between the real and forged images have become more pronounced. In an image, low-frequency regions typically represent semantic information, while high-frequency information describes edge and texture details. Unfortunately, neural networks often exhibit a bias towards fitting the low-frequency distribution during feature learning, thereby limiting the learning of high-frequency features. As a consequence, existing detection methods frequently associated with redundant parameters and insufficient information extraction. To address this issue, a multi-scale frequency decomposition module is devised to selectively extract valuable frequency features. This module is illustrated in Fig. 2.

Most of the frequency transformation involves a default operation of converting the image into a grayscale representation. Nonetheless, this operation may result in the loss of complex information contained in a single color channel. These channels may include crucial features that contribute to the distinction between real and fake images.

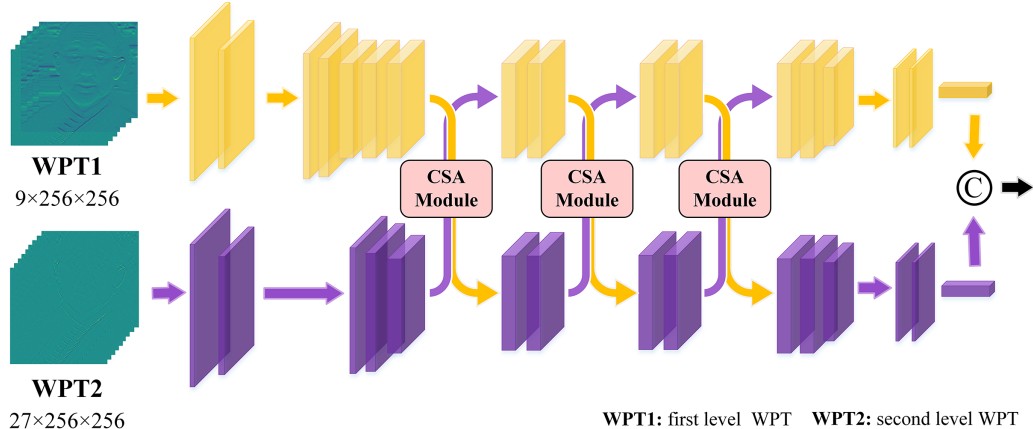

**Figure 2** **The structure of the multi-scale frequency decomposition module.**

Consequently, in the frequency branch, we retained the RGB channel, allowing for independent frequency transformation of the three channels. This approach ensures the retention of vital color information throughout the subsequent analysis.

The WPT is a widely employed method for extracting frequency information from an image. Unlike the discrete wavelet transform (DWT), which solely decomposes the low-frequency coefficients of the input data, WPT extends its capability by decomposing both low-frequency and high-frequency coefficients. By decomposing images into coefficients of varying resolutions and sizes, WPT facilitates the generation of diverse scale features, thereby enhancing the performance of low-resolution image detection.

In this work, we choose WPT to achieve effective decomposition of input images. Specifically, a two-level WPT is applied, resulting in the extraction of four distinct direction frequency coefficients. They are low-frequency coefficients, horizontal high-frequency coefficients, vertical high-frequency coefficients and diagonal high-frequency coefficients. These coefficients are obtained for each of the three independent color channels. Subsequently, the low-frequency components are discarded, and the first and second wavelet-packet coefficients are concatenated, forming nine and 27 channels respectively. These concatenated channels serve as inputs to the backbone network. Furthermore, the cross self-attention mechanism is employed to facilitate interaction and fusion of the multi-scale frequency features. We examined the efficiency between utilizing only high-frequency coefficients *vs.* using all frequency coefficients, more detailed information can be found in the "Experiments".

## Cross self-attention mechanism

Self-attention mechanism is an effective strategy which draws inspiration from human cognitive processes. The individuals can allocate their limited attention to the most relevant information during perception. Specifically, the self-attention method enables the model to learn the intrinsic relationships among different elements within the input. By computing an attention weight matrix, it establishes associations between each element

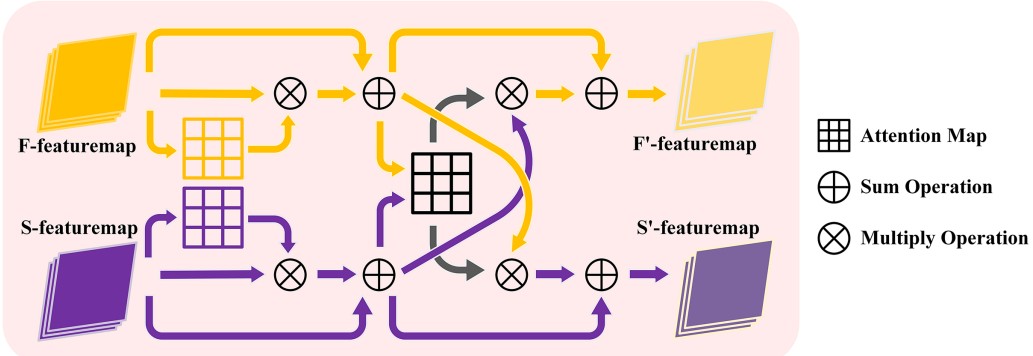

**Figure 3 The structure of cross self-attention module.**

and other parts of the input sequence. Therefore, self-attention is particularly adept at capturing dependencies within single-scale frequency information. Conversely, the cross attention facilitates the supervision of relationships between two distinct input sequences. Through the calculation of an attention weight matrix, it establishes correlations between each element of one sequence and all components of the other sequence. This characteristic enables the model to effectively connect and fuse the output coefficients derived from the first and second wavelet-packet transforms, thereby facilitating a deeper understanding of the interplay among multi-scale frequency features.

The proposed cross self-attention (CSA) module effectively combines the advantages of both attention strategy and multi-scale frequency features. Firstly, the frequency features from various scales are fed into a self-attention module, enabling the network to assign weights to the salient features within each scale's frequency information. These weighted frequency features are then outputted, and residual connections are employed to combine them with the initial features. Consequently, the multi-scale features are obtained through this process. During the learning process, three times CSA module are performed to facilitate the interaction and fusion of the multi-scale frequency features. The detailed process of this interaction is illustrated in Fig. 2 and the detailed operation is shown in Fig. 3. The CSA module is calculated as given in Eqs. (1)–(4).

$$CrossAtt = \text{soft}(\text{cat}(q_f, q_s)) \otimes \text{cat}(k_f, k_s) \tag{1}$$

$$F_i' = F_i \oplus (F_i \otimes SelfAtt(F_i)) \ \text{i} \in (f, s) \tag{2}$$

$$F_f'' = F_f' \oplus (F_s' \otimes CrossAtt) \tag{3}$$

$$F_s'' = F_s' \oplus (F_f' \otimes CrossAtt) \tag{4}$$

where $\otimes$ represents a matrix multiplication operation, and $\oplus$ represents an element addition operation. CrossAtt and SelfAtt means the cross attention and self-attention operation respectively. Equation (1) represents the calculation process of cross attention weight. $q_f$, $q_s$ represents the query of the features extracted by the network from the first and second wavelet coefficients, respectively. $k_f$ and $k_s$ represent the key of first and second level features respectively. $F_i'$ is the feature map calculated by self-attention. $F_f''$ and $F_s''$ shows the feature map processed by the cross attention.

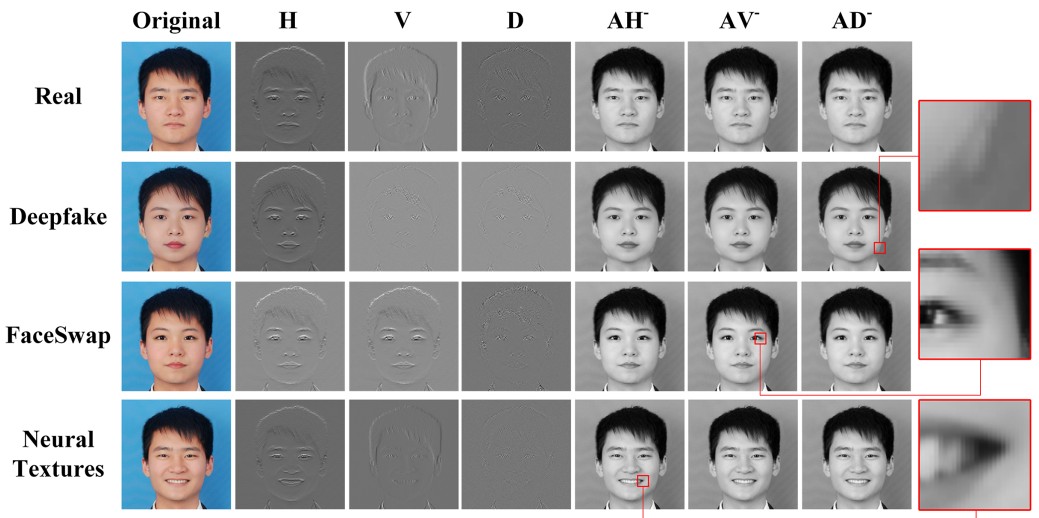

**Figure 4** **High-frequency component guided reconstruction images.**

## High frequency guided spatial feature extraction module

In the proposed frequency branch, high-frequency information can be represented better, while low-frequency information is more conducive to learn in the spatial domain. To address this, we introduce a multi-scale spatial feature extraction module aimed at capturing spatial features associated with high-frequency information. The WPT, employed in our approach, facilitates the extraction of spatial information across various directional scales. However, upon observation, we note that the spatial features corresponding to the high-frequency information present in individually inverted images are relatively minor. Consequently, it becomes challenging to discern image details in the resulting output, particularly after the inverse transformation of diagonal high-frequency components. To overcome this limitation, we ensure the preservation of low-frequency components during the WPT process. Additionally, we incorporate high-frequency information from three directions during the inverse transform. This approach not only retains the image's comprehensive information but also highlights manipulated areas with different directional scales in forged images.

To begin, we adopt a grayscale transformation for feature processing within the spatial branch. Subsequently, a standard wavelet-packet transform is applied to decompose the grayscale image into independent frequency coefficients. During the reconstruction phase, we focus on the three distinct high-frequency coefficients, while the unpaired points are filled with the zero matrix. This process can preserve semantic information while ensuring that each channel only retains components in a specific frequency domain. The specific steps involved in this process are visually depicted in Fig. 1C.

In Fig. 4, a visual analysis is shown to explain the reconstructed features in multi-scale spatial scales guided by high-frequency component. The first column is the original images. The two and four columns respectively display the high-frequency information of the image on the horizontal, vertical, and diagonal scales. It can be seen from the figure that

high-frequency information can carry the texture and edges information, which is crucial for distinguishing between true and fake images. This also explains our motivation for frequency feature selection. Columns 5 to 7 accordingly display the reconstruction images in different directions corresponding to high-frequency components. According to the enlarged images on the right, it can be observed that the reconstruction images contain different details in the spatial domain. After zooming in on the fifth column image, horizontal texture details can be seen. Similarly, vertical texture details can be observed in the enlarged image of the sixth column. In the last column of the image, diagonal textures can be observed.

## EXPERIMENTS

### Datasets and implement details

#### Datasets

In this work, we conduct our experiments on five publicly available face forgery detection datasets: FaceForensics++ (FF++) (*Rossler et al., 2019*), Celeb-DF (*Li et al., 2020b*), Deepfake Detection Challenge (DFDC) (*Dolhansky et al., 2020*), DFDC-Preview (DFDC-P) (*Dolhansky et al., 2019*) and DeepFakeDetection (DFD) (*Dufour & Gully, 2021*). These datasets provide videos with one real and the other deepfake counterparts.

*FaceForensics++.* The FaceForensics++ dataset contains 1,000 real videos extracted from Youtube. Fake videos are generated using both computer graphics-based and deep learning approaches (1,000 fake videos for each approach). The manipulation methods used in this dataset are Deepfakes (DF), Face2Face (F2F), FaceSwap (FS), and NeuralTextures (NT).

*Celeb-DF.* The Celeb-DF dataset aims to provide fake videos of better visual quality. It contains 890 real videos extracted from YouTube, corresponding to interviews of 59 celebrities with a diverse distribution in terms of gender, age, and ethnic group. As for fake videos, a total of 5,639 videos are created using a refined version of a public DeepFakes generation algorithm, which increasingly improves the synthesis quality.

*DFDC.* The DFDC dataset was released as part of a challenge. It consists of 124,000 video clips, with the real footage filmed by a wide range of actors under different scenarios and deepfaked with eight different techniques.

*DFDC-Preview.* DFDC-P dataset is a preview version of the DFDC dataset, which contains 5,214 videos and uses two forgery methods. In order to enhance the diversity of the dataset, careful consideration has been given to encompassing individuals of varying genders, ages, and skin tones. Moreover, the dataset encompasses a wide range of lighting conditions and head postures, while participants have utilized diverse backgrounds during the recording process.

*DeepFakeDetection.* The DFD dataset contains over 363 original sequences from 28 paid actors in 16 different scenes as well as over 3,000 manipulated videos using DeepFakes.

#### Evaluation metrics

We report the deepfake detection results with the most commonly used metrics in the literature, including the area under the receiver operating characteristic (ROC) curve (AUC) and accuracy (ACC). ACC can be described as the proportion of the correct

number of samples identified by the model to the total number. ACC and AUC are calculated as given in the Eqs. (5) and (6).

**TP (true positive):** Positive predictions that identified as true by the model.

**FP (false positive):** Negative predictions that erroneously identified as true by the model.

**TN (true negative):** Negative predictions that identified as false by the model.

**FN (false negative):** Positive predictions that erroneously identified as false by the model.

$$ACC = \frac{TP + TN}{TP + FN + FP + TN} \tag{5}$$

$$AUC = \frac{\sum_{i=1}^{P} rank_i - \frac{P(P+1)}{2}}{P \times N} \tag{6}$$

In the Eq. (6), $rank_i$ represents the predicted ranking of the $i$ positive sample, $P$ represents the number of positive samples, and $N$ represents the number of negative samples.

### Implement details

Regarding the experimental setup, this model was trained utilizing four NVIDIA TITAN V graphics processing units (GPUs) within a Linux-based environment. This experiment relies on the Python 3.10.8 environment and utilizes the PyTorch framework for both training and verification. Xception serves as the backbone network employed in this experiment. The network is initialized using pre-training weights obtained from the ImageNet dataset. The training batch size is set at 64. Adam optimizer is set with $\beta 1 = 0.9$, $\beta 2 = 0.999$, and $\varepsilon = 10^{-8}$. The learning rate is $2 \times 10^{-4}$.

## Experiment results and analysis

### Within dataset evaluation

In this section, we compare our method with current state-of-the-art deepfake detection methods on FF++ and DFDC. We first evaluate our methods on different video compression settings of FF++ including raw, high quality (HQ, C23), and low quality (LQ, C40). As the results shown in Table 2, the proposed method outperforms most of the methods in both ACC and AUC with all quality settings, especially in C23. This demonstrates our method' s detecting capabilities and anti-compression effect. Such an advantage would be impossible to accomplish without the proposed multi-scale spatial-frequency feature extraction network. By repeatedly filtering and reconstructing information at different scales, the model proposed can effectively distinguish suspicious information in forgery images. The incorporation of the CSA module enables the network to focus more precisely on significant frequency zones. The Multi-scale Patch Similarity Module proposed by local relation learning (LRL) (*Chen et al., 2021*) calculates the similarity between local regions through cosine distance, clearly modeling the relationships between different local regions. M2TR (*Wang et al., 2022*) utilizes the Transformer model

**Table 2 The performace of our method and other state-of-art methods on FaceForensics++ dataset.**

| Method | Raw | | C23 | | C40 | |
|---|---|---|---|---|---|---|
| | ACC | AUC (%) | ACC | AUC (%) | ACC | AUC (%) |
| Xception (*Chollet, 2017*) | 99.26 | 99.2 | 95.73 | 96.3 | 86.86 | 89.3 |
| Face X-ray (*Li et al., 2020a*) | – | – | – | 87.4 | – | 61.6 |
| $F^3$Net (*Qian et al., 2020*) | **99.95** | 99.8 | 97.52 | 98.1 | 90.43 | 93.3 |
| Two-branch (*Masi et al., 2020*) | – | – | 96.43 | 98.7 | 86.34 | 86.59 |
| WDB (*Jia et al., 2021*) | 99.74 | 99.78 | 96.95 | 99.6 | 88.96 | 92.97 |
| FDFL (*Li et al., 2021*) | – | – | 96.69 | 98.5 | 89.0 | 92.4 |
| LRL (*Chen et al., 2021*) | 99.87 | **99.92** | 97.59 | 99.46 | 91.47 | 95.21 |
| M2TR (*Wang et al., 2022*) | 99.50 | **99.92** | 97.93 | 99.51 | 92.89 | 95.31 |
| RECCE (*Cao et al., 2022*) | – | – | 97.06 | 99.32 | 91.03 | 95.02 |
| GocNet (*Guo et al., 2023c*) | – | – | 94.34 | 97.75 | 89.46 | 92.52 |
| LDFnet (*Guo et al., 2023b*) | – | – | 96.01 | 98.92 | 92.32 | **96.79** |
| Our | 99.62 | 99.87 | **97.98** | **99.64** | **92.92** | 94.35 |

Note:
Bold values refer to the best values.

**Table 3 The performace of our method and other state-of-art methods on DFDC dataset.**

| Method | ACC | AUC (%) |
|---|---|---|
| Xception (*Chollet, 2017*) | 89.83 | 94.86 |
| $F^3$Net (*Qian et al., 2020*) | 79.86 | 87.50 |
| M2TR (*Wang et al., 2022*) | 91.27 | 97.20 |
| GocNet (*Guo et al., 2023c*) | 92.52 | 96.87 |
| LDFnet (*Guo et al., 2023b*) | 93.15 | 97.20 |
| Our | **94.03** | **98.57** |

Note:
Bold values refer to the best values.

to perform self attention calculations on image blocks of different scales, capturing multi-scale artifacts. Both models have encoded masks and designed corresponding loss functions to guide the attention module in alleviating overfitting. We believe this is why these models can achieve better results in high-resolution data. Compared with M2TR, GocNet (*Guo et al., 2023c*) adopts tensor preprocessing module and manipulation tracking attention module, which further improves the detection performance of the backbone network while maintaining relatively low traffic. Although these works have achieved good detection results, their model parameters and computational costs are still relatively high.

Furthermore, we evaluate our method on the DFDC dataset, which is a more challenging task. We choose several state-of-the-art methods for a fair comparison, including $F^3$ Net (*Qian et al., 2020*), M2TR (*Wang et al., 2022*), GocNet (*Guo et al., 2023c*), LDFnet (*Guo et al., 2023b*). As shown in Table 3, our method outperforms other approaches by 0.88% and 1.37% in terms of ACC and AUC. These results validate the effectiveness of our proposed method under complicated scenarios.

**Table 4 Cross-dataset evaluation on Celeb-DF, DFDC, DFDCP and DFD dataset.**

| Method | Training | Test (AUC %) | | | |
|---|---|---|---|---|---|
| | | Celeb-DF | DFDC | DFDCP | DFD |
| Xception (*Chollet, 2017*) | FF++ | 65.36 | 67.9 | 72.2 | 70.5 |
| Two-branch (*Masi et al., 2020*) | | 73.41 | – | – | – |
| Face X-ray (*Li et al., 2020a*) | | 74.2 | 70.0 | 70.0 | 85.6 |
| F$^3$Net (*Qian et al., 2020*) | | 72.51 | 69.67 | – | 86.1 |
| Multi-attention (*Zhao et al., 2021a*) | | 76.54 | 67.36 | 66.28 | 75.53 |
| RECCE (*Cao et al., 2022*) | | **76.71** | 69.06 | – | – |
| GocNet (*Guo et al., 2023c*) | | 67.4 | – | – | – |
| LDFnet (*Guo et al., 2023b*) | | 65.7 | – | – | – |
| Our | | 75.73 | **70.48** | **73.41** | **89.26** |

**Note:**
Bold values refer to the best values.

### Cross-dataset evaluation

In this section, we evaluate the generalization ability of our method given that it is trained on FF++(raw) with multiple manipulations and tested on Celeb-DF, DFDC, DFDCP and DFD respectively. This settingis challenging in generalization ability evaluation since the testing sets are collected from different sources and share much less similarity with the training set. Table 4 analyzes the AUC of the proposed method with other current face forgery detection methods. Our method obtains 70.48%, 73.41% and 97.2% on DFDC, DFDCP and DFD respectively, which is outperformed other models. Especially compared to Xception (*Chollet, 2017*), our method has improved the detection performance on Celeb-DF by over 10%. Compared with F$^3$ Net (*Qian et al., 2020*), which also utilizes frequency information, our proposed method showed a higher AUC of 3.22% and 0.81% in generalization experiments on Celeb-DF and DFDC, respectively. The favorable generalization of this model is mainly attributed to the proposed multi-scale spatial feature extraction module. Thanks to the grouping reconstruction operation, the network can achieve more robust spatial feature extraction and obtain feature information from different directions. Meanwhile, the spatial branch effectively compensates for the limited feature learning and potential overfitting risks posed by relying just on frequency information.

It can be observed that multi-attention (*Zhao et al., 2021a*) has higher AUC on Celeb-DF dataset. It adopts a multi-level attention mechanism, relying on complex structures and powerful feature representation capabilities to learn rich common features, thus achieving better generalization. However, such large models often require significant computational resources and cannot achieve a good balance between speed and accuracy. In contrast, our proposed method can achieve comparable results in multiple cross-dataset experiments while maintaining the computational efficiency of the model.

### Ablation experiment

We first assess the efficacy of each module in our model, in which we develop the following experiment comparison on the DFDC dataset: (a) The baseline incorporates full-frequency

**Table 5 Ablation evaluation about effects of high-frequency filtering on DFDC dataset.**

| Method | | | DFDC | |
|---|---|---|---|---|
| | WPT1-A | WPT1 | ACC | AUC |
| a | ✓ | | 83.39 | 91.71 |
| b | | ✓ | **84.03** | **92.54** |

Note:
Bold values refer to the best values.

**Table 6 Ablation evaluation regarding wavelet-packet transformon on DFDC dataset.**

| | Method | | | DFDC | |
|---|---|---|---|---|---|
| | WPT1 | WPT2 | WPT3 | ACC | AUC |
| b | ✓ | | | 84.03 | 92.54 |
| c | ✓ | ✓ | | **87.29** | **94.6** |
| d | ✓ | ✓ | ✓ | 83.88 | 91.86 |

Note:
Bold values refer to the best values.

**Table 7 Ablation evaluation about the effect of CSA module and multi-scale spatial feature extraction module on DFDC dataset.**

| Method | | | | | DFDC | |
|---|---|---|---|---|---|---|
| | WPT1 | WPT2 | CSA | IWPT | ACC | AUC |
| c | ✓ | ✓ | | | 87.29 | 94.6 |
| e | ✓ | ✓ | ✓ | | 87.89 | 95.27 |
| f | ✓ | ✓ | ✓ | ✓ | **88.58** | **96.27** |

Note:
Bold values refer to the best values.

range information as input; (b) The baseline utilizes first level wavelet-packet transform to select high-frequency information as input; (c) The network incorporates a multi-scale frequency decomposition module to filter high-frequency information as input; (d) The model further employs a three-level wavelet-packet transform on top of the multi-scale frequency decomposition module; (e) The model incorporates both the multi-scale frequency decomposition module and the CSA module simultaneously; (f) The model combines the multi-scale frequency decomposition module, CSA module, and high-frequency guided spatial feature extraction module simultaneously. The quantitative results are listed in Tables 5–7.

By comparing a and b in Table 5, it can be concluded that retaining low-frequency information is not conducive to the effective learning of important features. This further demonstrates the necessity of feature selection for different frequencies.

Comparing variants b and c in Table 6, it can be concluded that the multi-scale frequency feature decomposition module can effectively obtain high-frequency information at different scales, helping the network improve detection accuracy. Method d illustrate that further use of the three-level wavelet-packet transform can result in more

pronounced noise in the image, with too little effective information, making it difficult to extract effective high-frequency information.

From variants e in Table 7, we observe an improvement on both ACC and AUC metrics when adding the CSA module which effectively achieve the interaction and fusion of high-frequency information at different levels, making the network more focused on the forged traces present in the frequency. By adding multi-scale spatial feature extraction module on top of e, more subtle artifacts in spatial information from different directions can learned by the model, which supplementing the network with necessary semantic and contextual information.The best performance is achieved when combining all the proposed components with ACC and AUC of 88.58% and 96.27% respectively.

## CONCLUSION

In this article, we propose a novel approach for face forgery detection by introducing a spatial and frequency dual stream network. The proposed model incorporates several modules that contribute to the effectiveness of the detection process. Specifically, the multi-scale frequency decomposition module effectively filters and extracts high-frequency features across multiple levels, enhancing the discriminative power of the network. Furthermore, the cross self-attention module facilitates the interaction and fusion of multi-scale frequency features, promoting a comprehensive analysis of the input data. Additionally, the high-frequency guided spatial feature extraction module employs grouping reconstruction techniques to enhance the model's robustness in extracting informative spatial features. The proposed method is extensively evaluated on widely-used benchmark datasets, which demonstrating its robustness and generalizability. Future work will focus on exploring the interactions between frequency and spatial domain information to address the challenges posed by complex real-world scenarios. This will allow for the development of more effective and reliable face forgery detection algorithms.

### Funding
This work was supported by the National Natural Science Foundation of China (Nos. 62006020, 62171042, 62102033). The funders had no role in study design, data collection and analysis, decision to publish, or preparation of the manuscript.

### Grant Disclosures
The following grant information was disclosed by the authors:
National Natural Science Foundation of China: 62006020, 62171042, 62102033.

### Competing Interests
The authors declare that they have no competing interests.

## Author Contributions

- Hui Zhao conceived and designed the experiments, performed the experiments, performed the computation work, prepared figures and/or tables, and approved the final draft.
- Xin Li conceived and designed the experiments, performed the experiments, performed the computation work, prepared figures and/or tables, and approved the final draft.
- Bingxin Xu conceived and designed the experiments, analyzed the data, authored or reviewed drafts of the article, and approved the final draft.
- Hongzhe Liu analyzed the data, authored or reviewed drafts of the article, and approved the final draft.

## Data Availability

The third-party public datasets are available at:

- FaceForensics++: https://github.com/ondyari/FaceForensics

- Celeb-DF: https://cse.buffalo.edu/~siweilyu/celeb-deepfakeforensics.html

- Deepfake Detection Challenge Dataset (DFDC) (8 facial forgery methods and124K videos) and Deepfake Detection Challenge Dataset (DFDC-P) (two facial forgery methods and 5K videos): https://ai.meta.com/datasets/dfdc/

The code is available at GitHub and Zenodo:

- https://github.com/RabbitMeet/train

- RabbitMeet. (2024). RabbitMeet/train: 1.0 (Version 1). Zenodo. https://doi.org/10.5281/zenodo.10462765.

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
