# Peer review of "Combined spatial and frequency dual stream network for face forgery detection"

_PeerJ Computer Science, doi:10.7717/peerj-cs.1959_

## Round 0.1 · original submission · Major Revisions

The representation of the work is not satisfactory and must be improved. There should be better linkage between sections and paragraphs, maintaining coherence throughout the paper. Additionally, the paper organization should be included at the end of the introduction section. It is suggested that the authors include a summary table for related work. Furthermore, the results should be explained in a more convincing manner.

Reviewer 1 ·

Basic reporting

This article proposes a combined spatial and frequency dual stream network for face forgery detection. Specifically, the multi-scale frequency decomposition module effectively filters and extracts high frequency features at different levels. The cross self-attention module can further promote the interaction and fusion of multi-scale frequency features.

Experimental design

More datasets needs to be included in the study.

Validity of the findings

More results on more datasets are required to validate the findings.

Additional comments

• Please highlight the advantages and disadvantages of your method.


• The authors should compare their methods with recent methods at the state of the art.

Cite this review as

·

Basic reporting

The overall structure is well organised. The introduction part reflects the necessary need to study fake face detection. The model part clearly represents the main feature extraction process and relationship of two proposed models. Experimental figures are relevant and well described to support the research work. From the perspective of improving this paper, the minor revisions are as follows:

Is there any recent research on using dual-stream network of spatial and frequency for fake face detection? It is suggested to further compare the differences between the existing closely related work and the author's network as well as the related advantages of the author's work in the introduction in the introduction part.

The English language should be improved, especially some prepositions such as comparison is generally used with or than instead of above. It is suggested that the authors revise the paper in terms of professional expressions.

Experimental design

In terms of experimental design, are the three datasets in the experiment representative? Are there other datasets of real scenarios in the field for testing?

Validity of the findings

Does the proposed network suffer from overfitting for face detection in real complex scenes, such as face angle and light occlusion? Please provide a detailed discussion on the generalisation and applicability of the proposed network, and specify the potential risk or feasible improvement direction of the proposed network.

Additional comments

no comment.

·

Basic reporting

Although this paper has a valuable effort and results, I have the following concerns :

1- There are some technical expressions that need to be clarified such as:
In the first sentence of the "Abstract", lines 11,12:
"With the development of generative model, the cost of facial manipulation and forgery is becoming lower
and lower "
>>>>>
With the development of the generative model, the cost of facial manipulation and forgery is becoming lower and lower. What is the meaning you wanted to convey to readers, and how does this meaning motivate you in this work?

In the midline 9 of the "Abstract" :
"Concretely, a multi-scale frequency attention (CSA)"
This is an incorrect acronym
The correct one you did later in the "Introduction", lines 52 and 53:
"Furthermore, a novel cross self-attention (CSA) module...."
Could we say: cross and self-attention or cross-self-attention?

These are just examples, please revise the whole paper in this regard.

2- The English language of this paper needs to be revised, especially the section entitled "Cross Self-attention Mechanism"

3- All Figures need to be of good quality and to have detailed explanations.

Experimental design

1- What are the limitations of the state-of-the-art, or what exactly is the research gap you want to fill?
2- Please give a step-by-step mechanism throughout the proposed structure.

Validity of the findings

I appreciate the experimental work done and the results obtained in this paper, however, I don't agree with the word "generalization" in the "Conclusion" section :
" ...,demonstrating its high accuracy and good generalization."

Based on the results shown in the Tables, your proposal does not always have better results.

Additional comments

Thanks and regards

Reviewer 4 ·

Basic reporting

The overall paper seems to be pretty easy to read but it could be improved in terms of presentation, structure and completeness. The experiments are not enough and their presentation is minimal with some unnecessary details/figures while missing of comments and deep explanations of the results.

Experimental design

I found a bit of confusion in the presentation of the metrics and variables e.g. AUC, TP and FN.
Even if these are pretty well-known concepts, the explanation by the authors is confusing. The

Implementation details are badly written with some useless information such as the OS.

In the tables in the experiment part, it would be good to see a comparison of computational resources used by the methods. Since the previous works reached very similar results with the proposed method, it would be useful to see if the result obtained by the authors is related to this aspect or not.

The overall experimental part is very limited. The authors just trained on one single dataset (FF++) and conducted in-dataset test on that data only. I appreciate the cross-dataset test on DFDC and Celeb-DF but I would like to see even just one additional experiment of training the model for example on DFDC and not only FF++. While another additional cross-dataset comparison could be using this exact same model trained on FF++, to test on DFDC Preview as done in many previous works. This is an easy experiment since does not require any additional training.

Figure 6 and Table 2 represent the exact same results, the figure is useless and repetitive and should be removed. The same for Figure 5 and Table 1 and for Figure 7 and Table 3. Why repeat all the results twice?


The captions of all the tables should be improved since they are too short and not enough informative.

Validity of the findings

Even if the results are comparable with previous works, the experiments are too few and the authors should try to improve the experimental part to strongly demonstrate their findings. Anyway, since the proposed method does not obtain significant improvements in state of the art but finish to be comparable with all the others, it would be useful to see how the proposed method perform on other dataset and see if in other contexts it achieve good performances.

Cite this review as

---

## Round 0.2 · accepted · Accept

Based on the reviewer’s recommendation, the paper is accepted for publication.

Reviewer 1 ·

Basic reporting

A cross self-attention (CSA)
module is designed to improve frequency feature interaction and fusion at different scales. The paper is interesting and apt.

Experimental design

Authors can include more technical discussion.

Validity of the findings

Yes, alright.

Additional comments

Authors may get the paper read by a fluent English speaker before publication.

Cite this review as

·

Basic reporting

All my previous concerns are well-addressed.

Experimental design

All my previous concerns are well-addressed.

Validity of the findings

All my previous concerns are well-addressed.

Additional comments

All my previous concerns are well-addressed.